# Overexpression of a Grape WRKY Transcription Factor *VhWRKY44* Improves the Resistance to Cold and Salt of *Arabidopsis thaliana*

**DOI:** 10.3390/ijms25137437

**Published:** 2024-07-06

**Authors:** Lihua Zhang, Liwei Xing, Jing Dai, Zhenghao Li, Aoning Zhang, Tianhe Wang, Wanda Liu, Xingguo Li, Deguo Han

**Affiliations:** 1Key Laboratory of Biology and Genetic Improvement of Horticultural Crops (Northeast Region), Ministry of Agriculture and Rural Afairs/National-Local Joint Engineering Research Center for Development and Utilization of Small Fruits in Cold Regions, College of Horticulture & Landscape Architecture, Northeast Agricultural University, Harbin 150030, China; zlh15009205209@163.com (L.Z.); xingliwei0413@126.com (L.X.); zzydzb@126.com (J.D.); lizhenghao1012@126.com (Z.L.); 17339831802@163.com (A.Z.); 2Horticulture Branch of Heilongjiang Academy of Agricultural Sciences, Harbin 150040, China; haaswth@126.com (T.W.); liuwanda@haas.cn (W.L.)

**Keywords:** grape rootstock, resistance gene, salinity and chilly stress, genetic transformation, transcriptional regulation

## Abstract

Plants are often exposed to biotic or abiotic stress, which can seriously impede their growth and development. In recent years, researchers have focused especially on the study of plant responses to biotic and abiotic stress. As one of the most widely planted grapevine rootstocks, ‘Beta’ has been extensively proven to be highly resistant to stress. However, further research is needed to understand the mechanisms of abiotic stress in ‘Beta’ rootstocks. In this study, we isolated and cloned a novel WRKY transcription factor, *VhWRKY44*, from the ‘Beta’ rootstock. Subcellular localization analysis revealed that VhWRKY44 was a nuclear-localized protein. Tissue-specific expression analysis indicated that *VhWRKY44* had higher expression levels in grape roots and mature leaves. Further research demonstrated that the expression level of *VhWRKY44* in grape roots and mature leaves was highly induced by salt and cold treatment. Compared with the control, *Arabidopsis* plants overexpressing *VhWRKY44* showed stronger resistance to salt and cold stress. The activities of superoxide dismutase (SOD), peroxidase (POD), and catalase (CAT) were significantly increased, and the contents of proline, malondialdehyde (MDA) and chlorophyll were changed considerably. In addition, significantly higher levels of stress-related genes were detected in the transgenic lines. The results indicated that *VhWRKY44* was an important transcription factor in ‘Beta’ with excellent salt and cold tolerance, providing a new foundation for abiotic stress research.

## 1. Introduction

Abiotic stresses and biological stresses such as pests and diseases limit the growth and geographical distribution of plants and have devastating effects on yield and fruit quality [1,2]. Cold and high salt stress are the two common abiotic environmental factors [3,4,5]. Cold and high salt stress can cause the metabolic imbalance of free radicals in cells, induce the accumulation of reactive oxygen species (ROS), destroy plant cell membrane, protease catalytic activity, and photosynthesis, ending with apoptosis under extreme circumstances [6,7]. Therefore, it is of vital significance to explore the mechanisms of plant responses to cold and salt stress and to mine related resistance genes and resistant materials.

To minimize the negative influences of abiotic stresses, plant cells have optimized complex defense mechanisms, which are mainly involved in the synthesis of protective compounds, such as proline, betaine, soluble sugars [8,9,10], antioxidant enzymes (SOD, POD, and CAT) and antioxidant substances (ascorbic acid, glutathione, flavonoids, etc.) that clear accumulated reactive oxygen species [11]. In addition, plant cells also give response via regulating gene expression. The present study revealed that plants respond to low-temperature stress through both CBF-dependent and non-CBF-dependent pathways and respond to salt stress with the salt overly sensitive (SOS), calcium-dependent protein kinase (CDPK) cascade reaction, phospholipids, mitogen-activated protein kinase (MAPK) cascade reaction, and abscisic acid (ABA) signaling pathway [12,13].

Transcription factors (TFs) can regulate the expression of related genes in response to stress to play a key role in response to various abiotic challenges [14,15,16]. Ethylene response factor *PtrERF108* activates the expression of the raffinose synthase gene *PtrRafS*, promoting the accumulation of raffinose to regulate the cold resistance of citrus [17]. Banana *MaNAC1* [18], apple *MdNAC029* [19], and *AP2/ERF* family transcription factor *MdABI4* [20] regulate plant cold resistance via CBF-dependent pathway, while *MdMYB88* improved the low temperature resistance of plant by activating the expression of *MdCBF3* or promoting anthocyanin accumulation [21]. For salt stress response, the allogeneic expression of *PvNAC52* improved the salt tolerance of *Arabidopsis* plants [22]. *OsMYBc* regulates *OsHKT1.1* expression in response to salt tolerance in rice, which is regulated by the ubiquitination of MYBc Stress-Related RING Finger Protein (MSRFP) [23]. *MdNAC047* regulates plant salt tolerance through ERF3-dependent ethylene pathway [24]. *MdMYB46* can enhance the tolerance of apple to salt stress by activating secondary cell wall biosynthesis pathway or directly activating stress response signals [25].

WRKY transcription factors are multifunctional proteins that regulate fruit ripening, senescence [26], and quality [27,28,29,30], and an increasing number of studies have shown that WRKY transcription factors in plants play a crucial regulatory role in single or even more complex environmental stresses [31,32]. *CsWRKY2*, which is involved in the ABA signaling pathway, significantly influences the response mechanism to cold and severe salt stress [33]. *TaWRKY33* and *TaWRKY1* were involved in the plant response to both drought stress and cold temperatures [34]. When *MBWRKY4* was in a high salinity environment and interacted with specific cis elements, it may have upregulated some of the downstream genes (*POD*, *APX*, *SOD*, etc.) associated with salt stress [35]. Furthermore, it has been demonstrated that *VvWRKY44* regulated grapevine stress in response to salt, drought, and cold temperatures [36]. In bananas, four WRKY regulators enhance the cold resistance of the fruit by regulating the ABA biosynthesis pathway [32]. The transcription factor *MdSPL13* targets the *MdWRKY100* gene, which enhances salt tolerance in apples [37]. The latest research shows that MdWRKY75-MdHSC70 modulates the heat resistance of apples via regulating the expression of heat shock factor genes [38]. Despite these advances, how WRKY transcription factors respond to complex abiotic stress processes remains to be investigated.

‘Beta’ is a remarkable hybrid of two grape varieties, *Vitis labrusca* and *Vitis riparia*, and serves as an excellent grape rootstock. It has proven to be exceptionally resistant to low temperature and has become a valuable germplasm resource [39]. This hybrid, well-known for its exceptional resistance to disease and cold, is a significant supporter of the local viticulture sector. Generally, rootstocks modify the nutrition and flavor of berries through their cation and free amino acid contents. However, ‘Beta’ has significantly improved fruit quality, sparking interest in its superior resistance and the pursuit of a premium fruit [40]. The high cold and salt tolerance of ‘Beta’ rootstocks will facilitate their steady expansion [41]. According to a recent study, WRKY44 is essential for plants to cope with abiotic stresses [39]. However, WRKY44 has not been explored in great detail in ‘Beta’. The aim of this study is to elaborate on the regulatory role of TFs, exploring the potential relevance of *WRKY* genes, and providing a precise framework for enhancing the salt and cold tolerance of grapevine rootstocks.

## 2. Results

### 2.1. VhWRKY44 Gene Cloning and Bioanalysis

The *VhWRKY44* gene is homologous to the ‘Pinot Noir’ (VIT_07s0005g01950.t01) gene. ProtParam was used to estimate the 478 amino acids and 1437 base pairs of the *VhWRKY44* CDs (Figure 1). The predicted molecular weight of the protein was approximately 52.37 kDa, and the calculated isoelectric point was 8.76. Significant amounts of Serine (Ser) (11.7%), Glutamate (Glu) (7.7%), Glycine (Gly) (7.7%), and Proline (Pro) (7.7%) are found in the amino acid fractions of the VhWRKY44 protein. The protein average hydrophilicity coefficient was estimated to be −0.988, indicating its high hydrophilicity. The protein was unstable, as indicated by the instability coefficient of 48.88.

The *VhWRKY44* gene contained two repetitive structural domains that belong to the WRKY family, similar to the other 11 members (Figure 2A). The created phylogenetic tree simultaneously depicted the evolutionary relationship between *VhWRKY44* and the other 11 species. *VhWRKY44* clustered with *VvWRKY44*, indicating the highest homology between these two genes, followed by *CsWRKY44*, *TcWRKY44*, *HuWRKY44*, and *AtWRKY44* (Figure 2B). The secondary structure of the VhWRKY44 protein consists of 75.94% random coils, 9.62% extended strands, 2.93% β-turns, and 11.51% α-helices (Figure 3A). *VhWRKY44* contained two WRKY domains located at amino acid positions 195–250 and 398–453, indicating that *VhWRKY44* was a member of the WRKY family (Figure 3B). In addition, the tertiary structure of VhWRKY44 protein was predicted using the SWISS-MODEL website, and its structure was consistent with the predicted secondary structure. (Figure 3C).

### 2.2. Localization of the VhWRKY44 Gene in Tobacco Cells

Meanwhile, we constructed the fusion vector *VhWRKY44*-pCAMBIA1300, which can be transiently expressed in tobacco, to investigate the specific subcellular localization of the *VhWRKY44* gene as a transcription factor. The *VhWRKY44* gene was introduced into tobacco leaves by the injection method. Subsequently, a laser confocal microscope was employed to observe the fluorescence dispersion in the tobacco leaves. The findings demonstrated that the nucleus of the tobacco infected with the recombinant plasmid exclusively displayed a blue color, while the fluorescence of the tobacco infected with the empty plasmid was distributed throughout the entire cell. Additionally, when the fluorescence of green fluorescent protein (GFP) and 4′,6-diamidino-2-phenylindole (DAPI) overlapped with the cell nucleus fluorescence, it was evident that *VhWRKY44* was located there (Figure 4).

### 2.3. Analysis of VhWRKY44 Expression Levels in ‘Beta’ Plants

In our experiments, we detected and analyzed the distribution of *VhWRKY44* expression levels in various ‘Beta’ tissues using RT-qPCR. The highest expression of *VhWRKY44* was observed in mature leaves, while roots, young leaves, and stems showed lower expression levels (Figure 5A).

*VhWRKY44* expression was induced under various stress conditions, including 200 mM NaCl, 4 °C, 6% PEG6000, 37 °C, and 100 μM ABA. Within 12 h of treatment, the expression level of *VhWRKY44* in mature leaves and roots showed similar peaks at different time intervals. In comparison, the peak expression of *VhWRKY44* in mature leaves and roots was 7.6 and 13.3 times higher, respectively, than that of the control group after 3 and 7 h of treatment with a high concentration of NaCl (Figure 5B). After 5 h and 3 h of treatment at 4 °C, respectively, the peak expression of *VhWRKY44* in mature leaves and roots was 10.45 times and 6.9 times higher than that of the blank control group (Figure 5C). The results revealed that the application of NaCl and 4 °C significantly increased the production levels of *VhWRKY44* in mature leaves and roots compared to other treatments.

### 2.4. Overexpression of VhWRKY44 Increase Salt Sensitivity of Arabidopsis thalian

Vector resistance screening was conducted on *VhWRKY44* overexpressed in the T_1_ generation of transgenic *Arabidopsis* and the null-loaded line (UL) to investigate the function of *VhWRKY44* during salt stress and cold stress. Transgenic *Arabidopsis* plants overexpressing *VhWRKY44* in the T_2_ generation were examined using RT-qPCR, with WT and UL plants serving as controls (Figure 6A). While the target gene could be consistently produced in transgenic *Arabidopsis*, these findings revealed that it was not expressed in the transgenic lines and showed variation in the control group. Finally, we screened three transgenic *Arabidopsis* lines (S2, S4, and S5) from a total of six transgenic *Arabidopsis* lines (S1–S6) with high expression levels and carefully cultured them until the T_3_ generation (Figure 6).

Experiments were conducted on WT, UL, and transgenic plants (S2, S4, and S5) to elucidate the role of *VhWRKY44* in *Arabidopsis* under severe salt stress conditions. WT, UL, and recombinant (S2, S4, and S5) plants were cultivated for four weeks at room temperature to observe plant phenotypes. Afterward, they were kept for 8 d in a nutrient dish containing a 200 mM NaCl solution. The growth rate of WT, UL, and transgenic plants (S2, S4, and S5) was similar under normal growth conditions (Figure 6B). When exposed to a 200 mM NaCl solution, the leaves of WT, UL, and transgenic (S2, S4, and S5) seedlings turned yellow. The leaves of the WT and UL plants withered and turned distinctly yellow (Figure 6B). The overexpression lines (S2, S4, and S5) exhibited survival rates of 88%, 92%, and 89%, respectively, following high salinity stress, while WT and UL had mortality percentages of 35% and 29%, respectively. This contrasts with the lines’ equal survival rates under control conditions (Figure 6C). In comparison to WT and UL, the overexpression lines showed a higher survival rate under high salt treatment, as indicated by the experimental results.

In both the control and high salinity treatments, we simultaneously assessed several important physiological indicators of WT and UL and S2, S4, and S5. The findings demonstrated that, under controlled circumstances, there was no discernible variation in the physiological index levels between the lines. However, the transgenic lines treated with high salt differed significantly from the WT and UL varieties in that they exhibited significantly higher levels of proline, SOD, POD, and CAT activities. They also exhibited significantly different levels of chlorophyll and MDA compared to the WT and UL lines (Figure 7A–F). Overexpressing *VhWRKY44* may enhance the resilience of transgenic plants to extreme salt stress. This demonstrates that the response of plants to extreme salt stress is significantly improved by overexpressing *VhWRKY44*.

### 2.5. VhWRKY44 Activates Genes Related to Salt Tolerance in Arabidopsis thaliana

To further clarify the regulatory mechanism of the *VhWRKY44* gene on salt stress, this experiment also investigated the expression of downstream-related response genes, including *AtNHX1* (NM_122597.3), *AtSOS1* (NM_126259.4), *AtSOS2* (NM_122932.5), *AtSOS3* (NM_122333.6), *AtCOR15a* (NM_129815.5), and *AtNCED3* (NM_112304.3), in *VhWRKY44* transgenic plants under salt stress conditions. The gene expression of *AtNHX1* (NM_122597.3), *AtSOS1* (NM_126259.4), *AtSOS2* (NM_122932.5), *AtSOS3* (NM_122333.6), *AtCOR15a* (NM_129815.5), and *AtNCED3* was detected using RT-qPCR analysis under normal conditions. Under conditions of extreme salt stress, the expression levels of these genes in the transgenic lines were noticeably higher than those in the WT and UL lines (Figure 8A–F). These modifications suggest that the *VhWRKY44* transgenic lines may enhance tolerance to severe salt stress by triggering the expression of genes associated with salt tolerance.

### 2.6. Overexpression of the VhWRKY44 Enhances Cold Resistance of Arabidopsis thaliana

We simultaneously confirmed the involvement of the *VhWRKY44* gene in cold tolerance using WT, UL, and transgenic *Arabidopsis* lines (S2, S4, and S5). The growth of each *Arabidopsis* line was robust, with no significant differences observed during the 4-week incubation period under normal seedling conditions. After a 12 h period of low temperatures at −4 °C, all *Arabidopsis* lines grew and developed slowly. Under the same conditions, both the WT and UL lines exhibited noticeable wilting, while only a small percentage of the genetically engineered lines (S2, S4, and S5) showed signs of wilting on their leaves (Figure 9A). After being exposed to normal temperature conditions and allowed to grow for one week, it was evident that the WT and UL lines had wilted leaves. At this point, 93.6%, 92.5%, and 88.3% of the leaves in the transgenic *Arabidopsis* lines overexpressing *VhWRKY44* (S2, S4, and S5) were able to grow normally, while the majority of plants in the WT and UL lines died (Figure 9B). *Arabidopsis* lines that overexpress *VhWRKY44* exhibited higher survival rates than WT and UL lines (UL) at a low temperature of −4 °C. This result suggests that the overexpression of *VhWRKY44* significantly enhances the survival of *Arabidopsis* in a cold environment.

Several physiological markers were also determined for WT, UL, and transgenic lines (S2, S4, S5) under normal conditions and after exposure to low temperature stress. The study’s findings demonstrated that the transgenic lines (S2, S4, and S5), as well as the WT and UL lines, exhibited significant variations in the concentrations of relevant physiological indicators after exposure to low temperatures. Specifically, the transgenic lines showed significantly increased SOD, POD, and CAT activities, as well as proline content, compared to the WT and UL lines. Significant differences in MDA and chlorophyll levels were observed simultaneously between the transgenic lines and the WT and UL lines (Figure 10A–F). This suggested that the overexpression of *VhWRKY44* can enhance the tolerance of transgenic plants to low temperature stress.

To gain a deeper understanding of the *VhWRKY44* gene’s regulatory mechanism in response to cold stress, we also investigated the expression levels of downstream response genes *AtKIN1* (NM_180411.3), *AtCBF1* (NM_118681.4), *AtCBF2* (NM_118679.2), *AtCBF3* (NM_118680.2), *AtCOR47* (NM_101894.4), and *AtRAB18* (NM_103516.4) in *VhWRKY44* transgenic plants following exposure to low temperatures. RT-qPCR analysis showed no significant differences in the transcription levels of the genes *AtCBF1* (NM_118681.4), *AtCBF2* (NM_118679.2), *AtCBF3* (NM_118680.2), *AtCOR47* (NM_101894.4), *AtKIN1* (NM_180411.3), and *AtRAB18* (NM_103516.4) in WT, UL, and transgenic lines under normal conditions. Following the low-temperature stress, the transgenic lines exhibited a much higher expression of the previously mentioned genes compared to the WT and UL lines (Figure 11A–F). This further demonstrates that the *VhWRKY44* transgenic lines may enhance tolerance to low temperature stress and increase the expression of genes associated with cold sensitivity.

## 3. Discussion

Low temperature and salt stress seriously damage plant growth and crop yield [1,2,5]. With the development of biotechnology, the breeding of resistant crops through transgenic methods has gradually matured. At present, some studies have demonstrated that WRKY family proteins play an essential role in abiotic stress response [31,32,33,34,35,36,37], while there are a few studies that have explored the anti-stress application of WRKY family proteins from resistant rootstocks ‘Beta’. In this study, *VhWRKY44* of ‘Beta’ was cloned and transferred into *Arabidopsis* seedlings to analyze its biological functions. Gene structure and phylogenetic analysis showed that *VhWRKY44* belonged to the WRKY family gene (Figure 2A). The changes in gene response to low temperature and salt stress and a series of related physiological indices in *VhWRKY44*-overexpressed *Arabidopsis* lines were further analyzed, revealing that *VhWRKY44* positively regulated the cold tolerance and salt tolerance of plants (Figure 12).

Abiotic stress can inhibit plant development and growth [42]. Under normal conditions, there was no significant growth phenotypic difference between *VhWRKY44*-overexpressed and the control *Arabidopsis* plants, while *VhWRKY44*-overexpressed seedlings maintained, respectively, better growth potential and higher survival rates compared to the control plants under the two stress conditions (Figure 6 and Figure 9), indicating that *VhWRKY44* can improve the adaptability of *Arabidopsis* plants to low temperature or salt stress. The accumulation of osmoregulatory substances such as proline can significantly reduce cell osmotic pressure, thus stabilizing membrane structure and normal function, and improving plant tolerance to abiotic stress [38,42]. Our results showed that the proline content of *VhWRKY44*-overexpressed *Arabidopsis* plants was significantly higher than that of control under low temperature or salt stress (Figure 7 and Figure 10), indicating that *VhWRKY44* might enhance the adaptability of plants to low temperature and salt damage by promoting the accumulation of osmoregulatory substances.

In addition, when plants are exposed to low-salt conditions for a long time, harmful substances MDA and reactive oxygen species (ROS) can accumulate excessively in the cells, which eventually leads to damage to the plants [43,44,45]. Antioxidant enzymes such as SOD, POD, and CAT are the main enzymes for ROS clearance, which can be detected under various abiotic stresses, such as drought [46,47,48,49], low temperature [50,51], and high salt [52] for accelerating ROS clearance, thereby maintaining plant cell homeostasis [45]. In this study, we found that the activities of antioxidant enzymes, i.e., SOD, POD, and CAT, were significantly higher than that of the control (Figure 7 and Figure 10), indicating that *VhWRKY44* could improve the activity of antioxidant enzymes and promote the clearance of excess ROS under stress induction.

Many studies have pointed out that WRKY transcription factors in plants play a crucial regulatory role in single or even more complex environmental stresses [31,32]. WRKY proteins can directly induce the expression of genes associated with cold tolerance in low-temperature environments. Low temperature and exogenous ABA treatments significantly induced *CsWRKY46* expression, and overexpressing *CsWRKY46* in *Arabidopsis* increased plant tolerance to low-temperature stress [53]. *OsWRKY71* overexpressed transgenic rice plants enhanced the expression of OsTGFR and WSI76 in response to low temperature stress, suggesting that *OsWRKY71* positively affects cold tolerance [54]. Additionally, WRKY TFs are essential for plant responses to salt stress. The *IbWRKY47* gene significantly enhanced sweet potato resistance to salt stress by regulating genes associated with salt stress [55]. Furthermore, *CmWRKY17,* as negative regulators, exhibited significantly lower salt stress responses compared to the control in chrysanthemum plants [56]. Therefore, the *VhWRKY44* gene identified in this study can improve the salt tolerance and low temperature tolerance of *Arabidopsis* plants at the same time (Figure 6 and Figure 9) and may play an important role in the process of plant response to complex abiotic stress.

The WRKY family is highly conserved in most plants throughout evolution and exclusively binds to the cis element W-box (TTGAC sequence) in the target gene’s promoter to regulate the target gene’s expression [57,58]. To analyze the possible downstream functional genes of *VhWRKY44* under low temperature and high salt stress, we measured the expression levels of key genes of the low salt stress response pathway [12,13] in transgenic plants under the two tested treatments, and the results showed that *VhWRKY44* was able to increase the expression level of *NCED3* via the ABA pathway and upregulate the effects of *SOS1*, *SOS2*, *SOS3*, and *NHX1* through the SOS pathway. This increased the delivery stage of *COR15a* in conjunction with the upstream target gene promoters. Furthermore, low-temperature stress also enhanced the transcription of downstream-related genes such as *KIN1*, *CBF1*, *CBF2*, *CBF3*, *COR47*, and *RAB18* (Figure 8 and Figure 11). Additionally, it was evident from the variations in the transcript levels of these downstream-related genes that *VhWRKY44* may enhance salt and cold tolerance.

## 4. Materials and Methods

### 4.1. Plant Materials and Treatment

A hybrid plant known as ‘Beta’ from Northeast Agricultural University in Harbin, China, was used in the current study. It is a hybrid of *Vitis labrusca* and *V. riparia*. The mature ‘Beta’ plant buds were sterilized and cultivated for 40 d in a mixture containing 0.2 mg/L of indole-3-butyric acid (IBA) (Solarbio, A8170, Beijing, China) and 0.5 mg/L of 6-aminopurine (6-BA) (Solarbio, A8170, Beijing, China). Once the well-grown plants reached medium root development (MS + 0.2 mg/L IBA + 0.1 mg/L 6-BA), the process of light culture was initiated. Cultivation was conducted in a laboratory light incubator (Jinghong GZP-350Y, Shanghai, China) at the College of Gardening and Environmental Architecture, Northeastern Agricultural College, with a temperature of 22 °C, 16 h of light, and 8 h of darkness. The seeds of the wild-type (WT) *A. thaliana* were selected from the Colombian ecotype, and the light culture conditions were consistent with those of the ‘Beta’ plants. The plants with strong root growth were transferred to the Hoagland nutrient solution for 7 d while maintaining normal growth conditions.

The expression of the *WRKY44* gene was analyzed in the roots, stems, and leaves (both young and old) of seventy ‘Beta’ seedlings. These seedlings had developed roots, stems, and leaves in anticipation of the upcoming stress treatment. The expression pattern of *VhWRKY44* may be determined by the following factors: drought, high levels of salt, low humidity, extreme temperatures, and ABA (Solarbio, A8060, Beijing, China) hormone stress. As seedlings in Hoagland’s solution, the plants were subjected to treatments including 200 mM NaCl (Solarbio, LA0200, Beijing, China), 4 °C, 37 °C, PEG6000 (Solarbio, P8250, Beijing, China), and 100 μM ABA. Samples of the stressed plants were taken at seven different time intervals (0, 1, 3, 5, 7, 9, and 12 h). The plants were immediately snap-frozen in liquid nitrogen and stored at a constant temperature of −80 °C in a refrigerator for further examination.

### 4.2. Cloning of VhWRKY44 Gene

Mature leaves of robust ‘Beta’ plants were used to extract total RNA. Subsequently, the first-strand cDNA was then synthesized using TransScript First Strand cDNA Synthesis SuperMix (TransGen, Biotech, Beijing, China). Using the SnapGene 6.0 software, we designed gene-specific primers (refer to Appendix A) based on the *VvWRKY44* nucleotide sequence in the grape (*Vitis vinifera* L.) gene (NM_001280988.1). The target gene was amplified using a BIO-RAD (Hercules, CA, USA) C1000^TM^ Touch Thermal Cycling PCR device. The target fragments that exhibited bright and distinct bands were purified using a 1% TAE solution, ligated into the T5 cloning vector, and then sequenced at UW Genetics in Beijing, China.

### 4.3. Subcellular Localization of VhWRKY44

The pCAMBIA1300-GFP plasmid (Shanghai BIOTECH Co., Ltd., Shanghai, China) and the full-length sequence of the *VhWRKY44* gene were utilized to generate subcellularly targeted expression vectors. To design specific primers, we utilized the cleavage sites *Sal* I and *BamH* I (Appendix A). To construct the *VhWRKY44*-GFP transient expression vector, the pCAMBIA1300-GFP vector was digested with the amplified *VhWRKY44* gene fragment. Afterward, the insert was ligated into the vector. At the same time, *Agrobacterium rhizogenes* transformation was conducted. The bacterial solution was activated, and the slime was collected and resuspended in a buffer containing 10 mM MgCl_2_, 10 mM MES, and 200 μM acetosyringone. The optical density (OD) value was adjusted to 0.4, and then, it was allowed to stand for two to three hours at room temperature. The infiltration solution was injected into the fully expanded leaves of Benjamin’s tobacco, which had been grown for 5–6 weeks, using a syringe. The cells were then left to incubate in low light for 48 h before being subjected to DAPI (10 μg/mL) staining. Tobacco cell nuclei were stained, and the stained leaves were examined under a laser confocal microscope (LSM 900, Precise, Beijing, China) to observe the intracellular distribution of proteins encoded by the *VhWRKY44* gene.

### 4.4. Sequence Alignment and Structure Analysis of VhWRKY44

The nucleotide sequence of *VhWRKY44* was aligned with *VvWRKY44* using the EMBOSS Needle tool available at https://www.ebi.ac.uk/Tools/psa/emboss_needle/ (accessed on 10 November 2023). Subsequently, the sequence was translated using the DNAMAN 6.0 software to predict the primary structure of VhWRKY44. The Expasy-ProtParam tool at https://web.expasy.org/protparam/ (accessed on 10 November 2023) was then utilized to predict the primary structure of the VhWRKY44 protein, including its relative molecular mass, theoretical isoelectric point, and average affinity coefficient. The structural domains and tertiary structure of the VhWRKY44 protein were predicted using the SWISS-MODEL (https://swissmodel.expasy.org/, accessed on 10 November 2023) and SMART (http://smart.embl-heidelberg.de/, accessed on 10 December 2023) protein structural domain analysis websites. The translated amino acid sequences were compared with WRKY sequences of other species with higher homology using BLAST on NCBI (https://blast.ncbi.nlm.nih.gov/Blast.cgi, accessed on 10 November 2023). Subsequently, the genealogical historical tree of the homologous proteins was constructed using the Mega7.0 software.

### 4.5. Expression Analysis of the VhWRKY44 Gene

Drawing on previous studies, we utilized a quantitative reverse transcription PCR (RT-qPCR) method to measure the expression level of the *VhWRKY44* gene. The internal reference gene selected for this study was the *Actin* gene (XM_002282480), which exhibited consistent expression. Additionally, highly specific RT-qPCR primers were designed based on the conserved regions of *Actin* and *VhWRKY44* (Appendix A). The TaKaRa SYBR^®^ Green Real-time PCR Master Mix from Beijing, China, was used in the RT-qPCR assay. First, pre-denaturation at 95 °C for 30 s, followed by re-denaturation at 95 °C for 5 s, and annealing at 60 °C for 10 s were the settings established for the quantitative reaction. This process was repeated 40 times. We employed the 2^−ΔΔCt^ method to examine the relative expression levels of the *VhWRKY44* gene [59]. Three technical replicates of the experiment were conducted.

### 4.6. Vector Preparation and Arabidopsis Plant Transformation

The homology arms and restriction sites (*BamH* I and *Sal* I) were created using the cloned coding sequence (CD) and the sequence of the pCambia1300 plasmid containing the CaMV35S promoter. By using homologous recombination primers and techniques, we created the overexpression vector *VhWRKY44*-Pcambia1300. The overexpression vector (*VhWRKY44*-pCAMBIA1300) and empty vector (pCAMBIA1300) were inserted into *Agrobacterium rhizogenes*, and then, the bacterial solution was transferred into *A. thaliana* by inflorescence infection. Using a medium containing 1/2 MS and 50 mg/L kana, *Arabidopsis* seeds were screened for positive seedlings in the T_1_ generation. When *Arabidopsis* plants reached maturity and seeds were collected for continued cultivation in the T_2_ generation, a quantitative analysis was conducted. Subsequently, T_3_ generation plants were grown, and high-expressing transgenic lines from the T_3_ generation (S2, S4, and S5) were selected to produce pure T_3_ plants.

### 4.7. Analysis of Stress-Related Physiological Indices in VhWRKY44-Overexpressed Arabidopsis Lines

The WT, the airborne line (UL), and the T_3_ generation transgenic *Arabidopsis* lines (S2, S4, S5) were cultured and grown for 4 weeks in a light incubator under the following conditions: temperature of 22 °C, humidity of 85%, and a light/dark cycle of 16 h/8 h. Each strain exhibiting good growth was chosen for salt stress (200 mM NaCl) and low-temperature stress (4 °C). The WT, UL, and T_3_ generation transgenic lines were irrigated with a salt solution daily for 8 d. The other robust lines were subjected to −4 °C treatment for 12 h and then returned to normal temperature conditions for one week of cultivation. During each treatment, each row was labeled accordingly, and the various states of the plants were recorded. Moreover, samples of *Arabidopsis* leaves from the WT, UL, and transgenic lines (S2, S4, S5) were collected to quantify MDA, chlorophyll (Chl), and the amino acid Pro using the thiobarbituric acid assay, a spectrophotometer, and an acid digestion method, respectively [60]. In a similar manner, UV spectrophotometry, the guaiacol method, and the tetranitrotetrazolium blue reduction method were employed to measure the activities of CAT, peroxidase POD, and SOD, respectively [61].

### 4.8. Analysis of Stress-Related Genes in VhWRKY44-Overexpressed Arabidopsis Lines

RNA was extracted and then converted into cDNA using a first strand as a template for WT, UL, and transgenic *Arabidopsis* after exposure to high salt, low temperature, and normal conditions (S2, S4, and S5). Relevant downstream genes of *VhWRKY44* were detected using RT-qPCR, including genes related to salt stress (*AtNHX1*, *AtSOS1*, *AtSOS2*, *AtSOS3*, *AtCOR15a*, and *AtNCED3*) and genes related to low temperature stress (*AtKIN1*, *AtCBF1*, *AtCBF2*, *AtCBF3*, *AtCOR47*, and *AtRAB18*). The table displays the specific primers used for the downstream response genes and internal reference genes (Appendix A).

### 4.9. Statistical Analysis

In this experiment, three biological replicates were conducted for each sample. Collection, processing, and determination of relevant indicators were performed. The experiment’s results are presented as the mean and the standard error (SE). An analysis of variance (ANOVA) was conducted using the SPSS 26.0 software to identify statistically significant differences (* *p* ≤ 0.05, ** *p* ≤ 0.01).

## 5. Conclusions

In this study, the researchers isolated and characterized the WRKY TF *VhWRKY44* in the grapevine rootstock betta (*Vitis labrusca* × *Vitis riparia*). Based on the results of subcellular localization studies, it was demonstrated that *VhWRKY44* was primarily concentrated in the nuclear region of the cells. According to the phylogenetic tree data, *VhWRKY44* and *VvWRKY44* were found to have the closest relationship. *VhWRKY44* was more strongly expressed in mature leaves and roots, and it showed greater responsiveness to signals of low temperature and high salt. The overexpression of *VhWRKY44* in *Arabidopsis* resulted in changes in related physiological parameters and an enhanced tolerance to high-salt and cold conditions. Similarly, it positively regulated the expression of salt stress-related genes (*NHX1*, *SOS1*, *SOS2*, *SOS3*, *COR15a*, *NCED3*) under high salt stress. Additionally, it promoted the expression of downstream genes (*KIN1*, *CBF1*, *CBF2*, *CBF3*, *COR47*, *RAB18*) associated with low-temperature stress. The study results indicated that overexpressing *VhWRKY44* was crucial for enhancing plants’ ability to withstand cold and salt. This provides a basis for molecular breeding improvements in grape rootstocks.

## Figures and Tables

**Figure 1 ijms-25-07437-f001:**
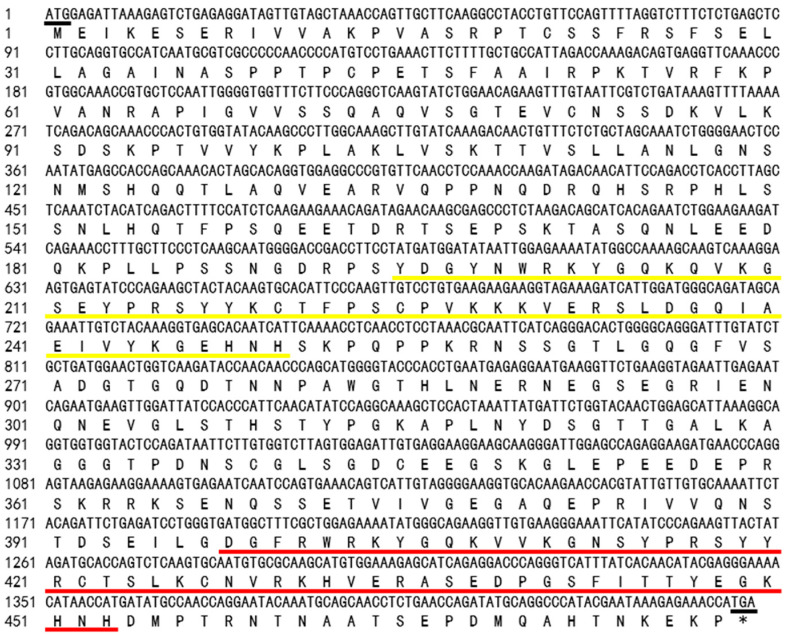
Nucleotide and deduced amino acid sequence of VhWRKY44. Black underlining indicates the start codon and the stop codon, yellow and red underlining indicate the two conserved structural domains, respectively. * stands for termination codon.

**Figure 2 ijms-25-07437-f002:**
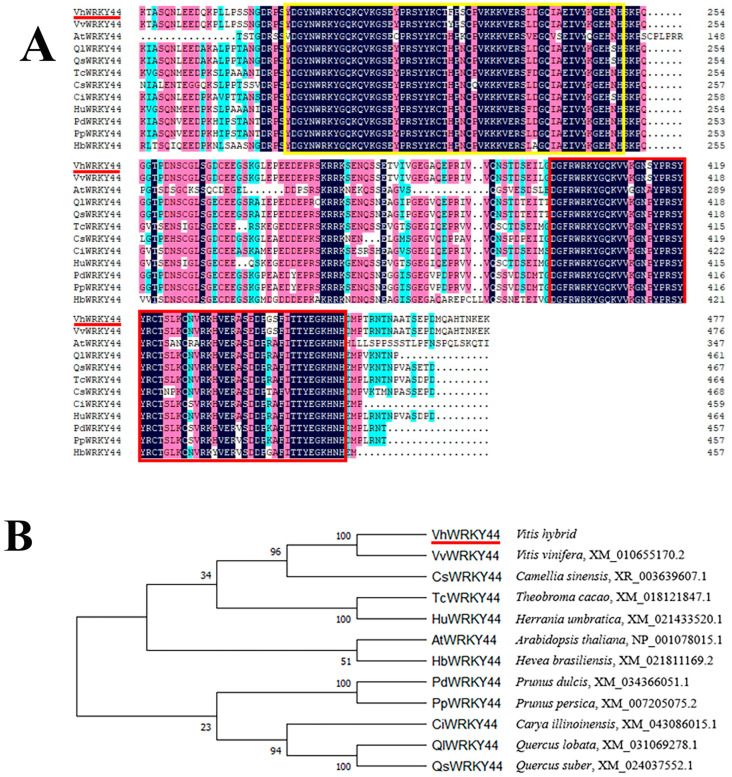
Comparative relationship between VhWRKY44 and WRKY44 proteins of other species. (**A**) VhWRKY44 in comparison to other species’ WRKY44 proteins. Yellow and red boxes indicate two WRKY conserved structural domains. (**B**) WRKY44 and VhWRKY44 protein phylogenetic trees of different species. The protein VhWRKY44 is shown by the red underline.

**Figure 3 ijms-25-07437-f003:**
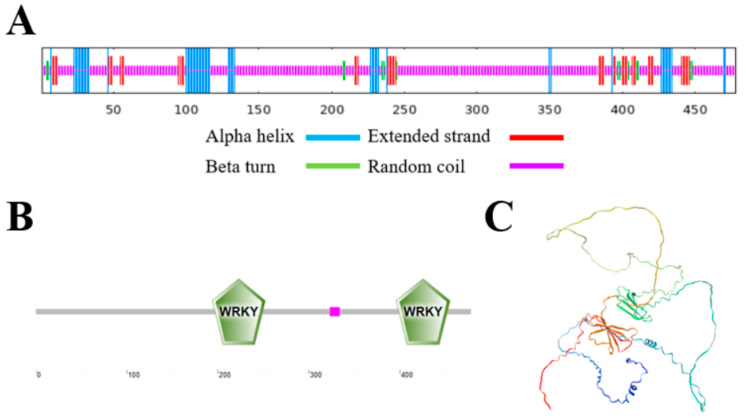
Structural examination of the protein VhWRKY44. (**A**) VhWRKY44′s structural analysis at the secondary level. (**B**) Conserved structural domain study of VhWRKY44. (**C**) Tertiary structural prediction of VhWRKY44.

**Figure 4 ijms-25-07437-f004:**
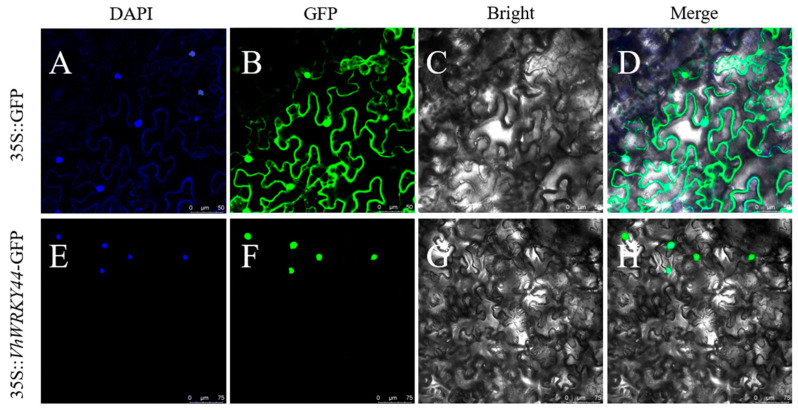
VhWRKY44 protein subcellular distribution in tobacco cells. (**A**–**D**) Where the vacant vector 35S::GFP is located in tobacco cells in the epidermis, bar = 50 μm. (**E**–**H**) Recombinant vector 35S::*VhWRKY44*-GFP localization in tobacco, bar = 75 μm. Images of DAPI, GFP, Bright, and Merge are shown from left to right.

**Figure 5 ijms-25-07437-f005:**
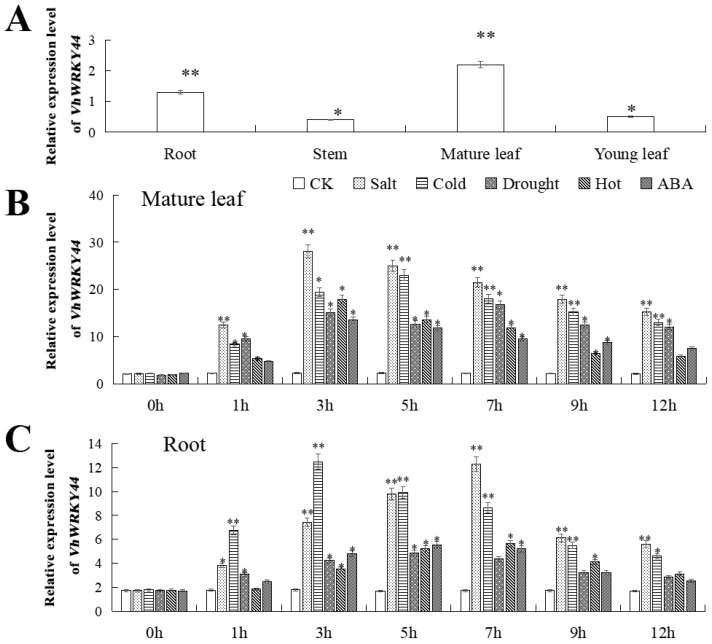
Expression of *VhWRKY44* gene in ‘Beta’. (**A**) Roots, stems, mature leaves, and young leaves all exhibit varying degrees of *VhWRKY44* expression. (**B**) Variation in *VhWRKY44* expression in fully grown leaves following 200 mM NaCl, 4 °C, 6% PEG6000, 37 °C, and 100 μM ABA treatments, in that order. (**C**) Roots treated with 200 mM NaCl, 4 °C, 6% PEG6000, 37 °C, and 100 μM ABA showed varying levels of *VhWRKY44* expression. These data are averages from three replicates. An asterisk appears above the error bars to indicate whether there is an important distinction among the treatment and control groups (* *p* ≤ 0.05; ** *p* ≤ 0.01).

**Figure 6 ijms-25-07437-f006:**
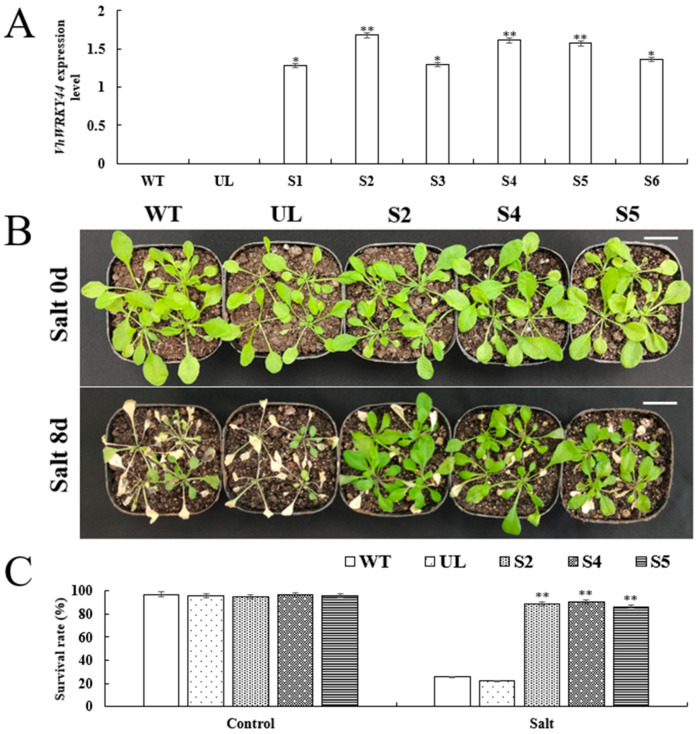
35S::*VhWRKY44 Arabidopsis* transgenic plants’ ability to withstand salt. (**A**) Expression of the *VhWRKY44* transcript in transgenic lines S1–S8, null-loaded line UL, and wild-type WT. (**B**) Transgenic high-expression lines (S2, S4, and S5), UL, and WT phenotypes. Scale bar is 3 cm. (**C**) Survival of each *Arabidopsis* line under high salt stress. Asterisks above indicate statistically significant comparisons (* *p* ≤ 0.05; ** *p* ≤ 0.01) with WT differences.

**Figure 7 ijms-25-07437-f007:**
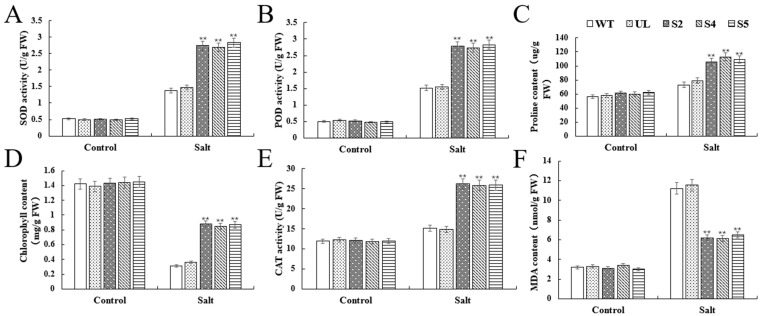
Effects of WT, UL, and 35S::*VhWRKY44* transgenic *Arabidopsis* lines on physiological indices under high salt stress: (**A**) SOD activity; (**B**) POD activity; (**C**) proline content; (**D**) chlorophyll content; (**E**) CAT activity; and (**F**) MDA content. The mean of three replicates is the standard error. The transgenic lines differed significantly from the WT lines, as indicated by the asterisk above the error bars (** *p* ≤ 0.01).

**Figure 8 ijms-25-07437-f008:**
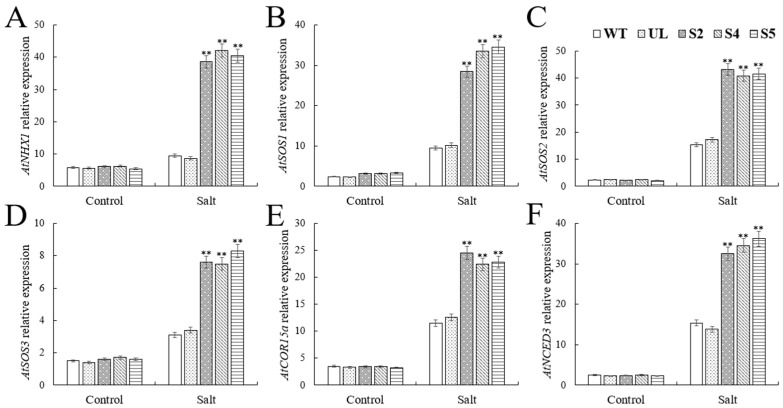
Using the RT-qPCR method, the expression of six genes associated with salt tolerance in WT and transgenic *Arabidopsis* under high salt stress was examined: (**A**) *AtNHX1*; (**B**) *AtSOS1*; (**C**) *AtSOS2*; (**D**) *AtSOS3*; (**E**) *AtCOR15a*, and (**F**) *AtNCED3.* The mean of three duplicate experiments is the standard error. The transgenic lines differed significantly from the WT lines, as indicated by the asterisk above the error bars (** *p* ≤ 0.01).

**Figure 9 ijms-25-07437-f009:**
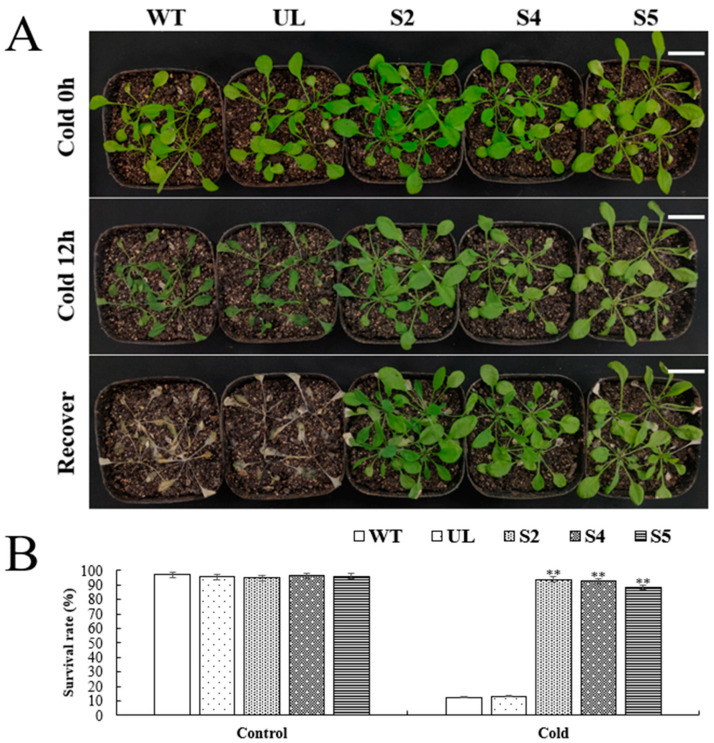
Cold tolerance of 35S::*VhWRKY44* genetically engineered *Arabidopsis* seedlings. (**A**) The characteristics of WT, UL, and recombinant high-expression lines (S2, S4, and S5). Scale bar is 3 cm. (**B**) Survival of each *Arabidopsis* line under low-temperature stress. Asterisks above indicate statistically significant comparisons (** *p* ≤ 0.01) with WT differences.

**Figure 10 ijms-25-07437-f010:**
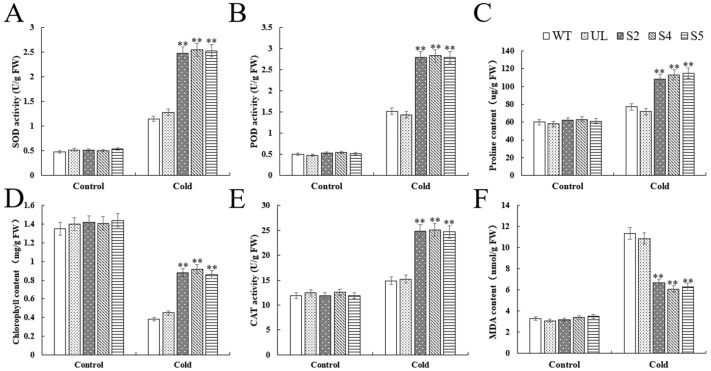
Effects of WT, UL, and 35S::*VhWRKY44* transgenic *Arabidopsis* lines on physiological indices under low temperature stress: (**A**) SOD activity; (**B**) POD activity; (**C**) proline content; (**D**) chlorophyll content; (**E**) CAT activity; and (**F**) MDA content. The mean of three replicates is the standard error. The transgenic line differs considerably from the WT line, as indicated by the asterisk above the error bars (** *p* ≤ 0.01).

**Figure 11 ijms-25-07437-f011:**
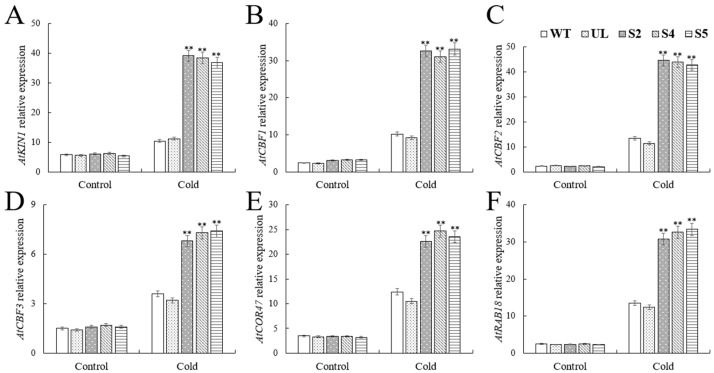
Using the RT-qPCR method, the expression of six genes associated with cold stress was examined in WT and transgenic *Arabidopsis* during low temperature stress: (**A**) *AtKIN1*; (**B**) *AtCBF1*; (**C**) *AtCBF2*; (**D**) *AtCBF3*; (**E**) *AtCOR47*, and (**F**) *AtRAB18*. Standard error is the mean of 3 replicates of the experiments. An asterisk above the error bars indicates that the transgenic lines differed significantly from the WT lines (** *p* ≤ 0.01).

**Figure 12 ijms-25-07437-f012:**
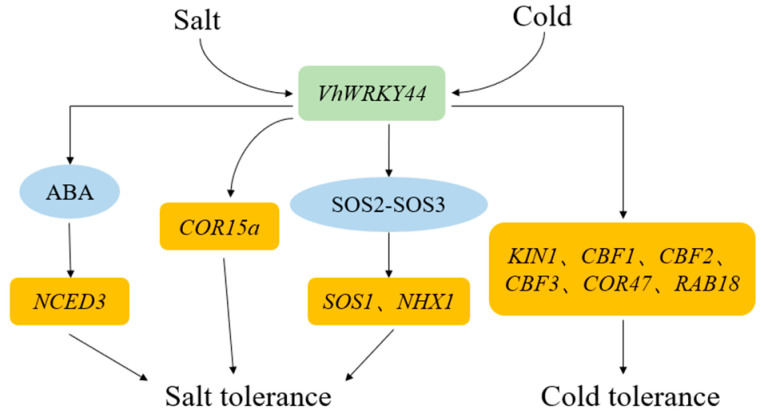
Potential models for the *VhWRKY44* gene following salt stress and low temperature stress. NCED3, Nine-Cis-Epoxycarotenodi Dioxyg-Enase 3; COR15a, Cold-responsive 15a; NHX1, Na+/H+ Antiporter 1; SOS1, Salt Overly Sensitive 1; SOS2, Salt Overly Sensitive 2; SOS3, Salt Overly Sensitive 3; KIN1, Cold-induced Protein 1; CBF1, C-repeat Binding Factor 1; CBF2, C-repeat Binding Factor 2; CBF3, C-repeat Binding Factor 3; COR47, Cold-regulated 47; and RAB18, Responsive To ABA 18.

## Data Availability

Data is contained within the article or Appendix A.

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
