# Peer review of "Overexpression of a Grape WRKY Transcription Factor VhWRKY44 Improves the Resistance to Cold and Salt of Arabidopsis thaliana"

_ijms, 2024, doi:10.3390/ijms25137437_

Round 1
Reviewer 1 Report
Comments and Suggestions for Authors
Though the work is interesting and could be considered for publication in IJMS. However, the current version needs to be further improved. Below are some suggestions:
1. The keywords should not be similar to the title. Please consider using different keywords.
2. In the main text, the citation numbers are confusing as part of the text. Please carefully format the citation according to the journal guidelines. Or atleast make them isolated from the main text.
3. Lines 32-36, and 63-67 are quite similar.
1. Please start a new paragraph from line 63 and these lines should not be similar to the above lines. The authors need to provide some recent citations for line 63-72, which are about cold and salt stresses, such as doi; 10.1111/ppl.14188, 10.1146/annurev-arplant-061422-104322, etc. The same references can also be used in the discussion to reduce the number of references.
2. Lines 80-82, the authors repeat the same what is already discussed in the above two paragraphs. The authors should carefully work and concise the introduction with the most suitable information.
3. Table 1 can be moved to a supplementary table. It is not important in the main text.
4. Fig 3A, remove Chinese characters.
5. The discussion is also too long and somewhat redundant to the introduction. Please improve it.
Author Response
Comments and Suggestions for Authors:
Though the work is interesting and could be considered for publication in IJMS. However, the current version needs to be further improved. Below are some suggestions:
Q1. The keywords should not be similar to the title. Please consider using different keywords.
Response: Thanks for your comments. We have revised it. Please check line30-31.
Q2. In the main text, the citation numbers are confusing as part of the text. Please carefully format the citation according to the journal guidelines. Or atleast make them isolated from the main text.
Response: We have checked the whole manuscript and revised it. Please check it.
Q3. Lines 32-36, and 63-67 are quite similar.
Response: Based on the reviewer's suggestions, we have made major modifications in the introduction part and added some latest references according to the reviewer's suggestions. Please check the introduction part
Q4. Please start a new paragraph from line 63 and these lines should not be similar to the above lines. The authors need to provide some recent citations for line 63-72, which are about cold and salt stresses, such as doi; 10.1111/ppl.14188, 10.1146/annurev-arplant-061422-104322, etc. The same references can also be used in the discussion to reduce the number of references.
Response: Based on the reviewer's suggestions, we have made major modifications in the introduction part and added some latest references according to the reviewer's suggestions. Please check the introduction part
Q5. Lines 80-82, the authors repeat the same what is already discussed in the above two paragraphs. The authors should carefully work and concise the introduction with the most suitable information.
Response: Based on the reviewer's suggestions, we have made major modifications in the introduction part and added some latest references according to the reviewer's suggestions. Please check the introduction part.
Q6. Table 1 can be moved to a supplementary table. It is not important in the main text.
Response: We have revised it and moved it to supplementary table 1. Please check it.
Q7. Fig 3A, remove Chinese characters.
Response: We have revised it.
Q8. The discussion is also too long and somewhat redundant to the introduction. Please improve it.
Response: Thanks for your professional suggestion. We have made major modifications in this part, please check it.
Reviewer 2 Report
Comments and Suggestions for Authors
The manuscript ‘Overexpression of a grape WRKY transcription factor VhWRKY44 improves the resistance to cold and salt of Arabidopsis thaliana’ by Zhang et al., is an interesting research that has been fairly well constructed. The obtained data and their experimental findings deserve to be published. I only have some minor point to make as flowing. I hope the authors find them useful.
Line 24: change ‘stressors’ to ‘’stress’’
Line 25 add ‘’enzymes’’ after ‘CAT’
Line 29: change the keywords to words that are not already in the manuscript title. For example change ‘salt’ to ‘’salinity stress’’
Line 33 and elsewhere: Put the reference numbers in brackets []
Line 37: change the ‘expansion’ to a better term
Line 37: change ‘to play a key role’ to ‘’and play key role’’
Line 45: remove the ‘, which’ and give a reference for this statement
Line 54: give a references
Line 56: reference for ‘related research’
Line 60: change ‘gene AtWRKY46’ to ‘’ AtWRKY46 gene’’
Line 70: reference for the sentences ending with ‘…apoptosis.’
Line 100: change ‘2628’ to ‘’[26-28]’’
Line 103: ‘resistant’ to what?
Line 104: this sentence can be merged with the earlier and shorten
Line 111: always keep the variety name as "Beta"
Table 1: adjust the primer sequences to be distinguishable and add the accession numbers when a single gene is referred to for qPCR
Table 1: you can underline the cleavage sites of Sal I and BamH I in table 1
Line 145: italicize ‘Agrobacterium rhizogenes’
Lin2 216: bring some more references for the applied assays such as MDA
Fig 5: I suggest to insert the VhWRKY44 name in the charts somewhere that can be interpreted even without the caption
Line 391 and elsewhere: I suggest to change the ‘strains’ to ‘’lines’’
Line 430-450: I suggest removing or minimizing the redundant information that was already mentioned in the introduction. The discussion needs more sever revision. I suggest the authors to avoid the repletion of the applied methods in this section and to compare and discuss their main findings in the context plant stress physiology.
Line 557: give the full name of the abbreviations in figure caption.
Author Response
Comments and Suggestions for Authors:
The manuscript ‘Overexpression of a grape WRKY transcription factor VhWRKY44 improves the resistance to cold and salt of Arabidopsis thaliana’ by Zhang et al., is an interesting research that has been fairly well constructed. The obtained data and their experimental findings deserve to be published. I only have some minor point to make as flowing. I hope the authors find them useful.
Q1. Line 24: change ‘stressors’ to ‘stress’
Response: Thanks a lot. We have revised it. Please check line25.
Q2. Line 25 add ‘enzymes’ after ‘CAT’
Response: We have revised it. Please check line25.
Q3. Line 29: change the keywords to words that are not already in the manuscript title. For example change ‘salt’ to ‘’salinity stress’’
Response: We have revised it. Please check line30-31.
Q4. Line 33 and elsewhere: Put the reference numbers in brackets.
Response: We have checked the whole manuscript and revised it. Please check it.
Q5. Line 37: change the ‘expansion’ to a better term
Response: Based on the reviewer's suggestions, we have made major modifications to this introduction part. Please check it.
Q6. Line 37: change ‘to play a key role’ to ‘’and play key role’’
Response: We have revised it. Please check line 55.
Q7. Line 45: remove the ‘, which’ and give a reference for this statement
Response: We have made a number of changes to the introduction and discussion sections, adding logic and readability of this article, and have removed this sentence detailing the WRKY domain.
Q8. Line 54: give a references
Response: We have revised it. Plaease check line 54-55.
Q9. Line 56: reference for ‘related research’
Response: We have made a number of changes to the introduction and discussion sections, adding logic and readability of this article, and have removed this sentence detailing the WRKY domain.
Q10. Line 60: change ‘gene AtWRKY46’ to ‘’ AtWRKY46 gene’’
Response: We have revised it.
Q11. Line 70: reference for the sentences ending with ‘…apoptosis.’
Response: We have revised it. Plaease check line 39
Q12. Line 100: change ‘2628’ to ‘’[26-28]’’
Response: We have checked the whole manuscript and revised it. Please check it.
Q13. Line 103: ‘resistant’ to what?
Response: Based on the reviewer's suggestions, we have made major modifications in the introduction part and added some latest references according to the reviewer's suggestions. Please check the introduction part.
Q14. Line 104: this sentence can be merged with the earlier and shorten
Response: Based on the reviewer's suggestions, we have made major modifications in the introduction part and added some latest references according to the reviewer's suggestions. Please check the introduction part.
Q15. Line 111: always keep the variety name as "Beta"
Response: We have checked the whole manuscript and revised it.
Q16. Table 1: adjust the primer sequences to be distinguishable and add the accession numbers when a single gene is referred to for qPCR
Response: We have revised it and moved it to supplementary table 1. Please check it.
Q17. Table 1: you can underline the cleavage sites of Sal I and BamH I in table 1
Response: We have revised it and moved it to supplementary table 1. Please check it.
Q18. Line 145: italicize ‘Agrobacterium rhizogenes’
Response: We have checked the whole manuscript and revised it. Please check line 409 and 451.
Q19. Lin 216: bring some more references for the applied assays such as MDA
Response: Thanks a lot. The determination method is based on previous studies in our laboratory.
Q20. Fig 5: I suggest to insert the VhWRKY44 name in the charts somewhere that can be interpreted even without the caption
Response: We have revised it. Please check Fig 5.
Q21. Line 391 and elsewhere: I suggest to change the ‘strains’ to ‘’lines’’
Response: We have checked the whole manuscript and revised it. Please check line 460.
Q22. Line 430-450: I suggest removing or minimizing the redundant information that was already mentioned in the introduction. The discussion needs more sever revision. I suggest the authors to avoid the repletion of the applied methods in this section and to compare and discuss their main findings in the context plant stress physiology.
Response: Based on the reviewer's suggestions, we have made major modifications in the discussion part. Please check it.
Q23. Line 557: give the full name of the abbreviations in figure caption.
Response: We have revised it. Please check Fig 12.
Round 2
Reviewer 1 Report
Comments and Suggestions for Authors
Though some of the comments have been addressed, some suggestions are untouched. See below.
1. The keywords should not be similar to the title. Please consider using different keywords. Four keywords are still similar to the title. For high-impact indexing and keyword searches, please use some new keywords that have not been used in the title.
2. Line 35, 60, 88, 89 and so on, fix the errors. I don’t know if the authors checked the draft before submission.
3. Start a new paragraph from line 63, and these lines should not be similar to the above lines. The authors need to provide some recent citations for line 63-72, which are about cold and salt stresses, such as doi; 10.1111/ppl.14188, 10.1146/annurev-arplant-061422-104322, etc. The same references can also be used in the discussion to reduce the number of references……Though the text has been slightly improved but, they did not make an effort to incorporate the cold stress-reference-related suggestions (doi; 10.1111/ppl.14188,). I cannot see any update in the references as suggested.
4. Again, authors need to carefully check the whole text before resubmission.
Author Response
- The keywords should not be similar to the title. Please consider using different keywords. Four keywords are still similar to the title. For high-impact indexing and keyword searches, please use some new keywords that have not been used in the title.
Response: Thanks for your comments. We have revised it. Please check line30-31.
- Line 35, 60, 88, 89 and so on, fix the errors. I don’t know if the authors checked the draft before submission.
Response:We have revised them, thanks.
- Start a new paragraph from line 63, and these lines should not be similar to the above lines. The authors need to provide some recent citations for line 63-72, which are about cold and salt stresses, such as doi; 10.1111/ppl. 14188, 10.1146/annurev-arplant-061422-104322, etc. The same references can also be used in the discussion to reduce the number of references……Though the text has been slightly improved but, they did not make an effort to incorporate the cold stress-reference-related suggestions (doi; 10.1111/ppl.14188,). I cannot see any update in the references as suggested.
Response:We have revised it, added relevant citations, and used green font for the modified part. Please check it.
- Again, authors need to carefully check the whole text before resubmission.
Response: We have checked the full text carefully.
Round 3
Reviewer 1 Report
Comments and Suggestions for Authors
Now it can be accepted.